# Unveiling the Oxazolidine Character of Pseudoproline Derivatives by Automated Flow Peptide Chemistry

**DOI:** 10.3390/ijms25084150

**Published:** 2024-04-09

**Authors:** Szebasztián Szaniszló, Antal Csámpai, Dániel Horváth, Richárd Tomecz, Viktor Farkas, András Perczel

**Affiliations:** 1Laboratory of Structural Chemistry and Biology, Institute of Chemistry, ELTE Eötvös Loránd University, Pázmány Péter Sétány 1/A, 1117 Budapest, Hungary; szaniszlo.szebasztian@ttk.elte.hu (S.S.); daniel.horvath@ttk.elte.hu (D.H.); tomecz.richard@gmail.com (R.T.); 2ELTE Hevesy György Ph.D. School of Chemistry, ELTE Eötvös Loránd University, Pázmány Péter Sétány 1/A, 1117 Budapest, Hungary; 3Instutite of Chemistry, ELTE Eötvös Loránd University, Pázmány Péter Sétány 1/A, 1117 Budapest, Hungary; antal.csampai@ttk.elte.hu; 4HUN-REN—ELTE Protein Modeling Research Group, ELTE Eötvös Loránd University, Pázmány Péter Sétány 1/A, 1117 Budapest, Hungary

**Keywords:** pseudoproline, automated flow peptide synthesis, oxazolidine ring opening, aspartimide

## Abstract

Pseudoproline derivatives such as Thr(ΨPro)-OH are commonly used in peptide synthesis to reduce the likelihood of peptide aggregation and to prevent aspartimide (Asi) formation during the synthesis process. In this study, we investigate notable by-products such as aspartimide formation and an imine derivative of the Thr(ΨPro) moiety observed in flow peptide chemistry synthesis. To gain insight into the formation of these unexpected by-products, we design a series of experiments. Furthermore, we demonstrate the oxazolidine character of the pseudoproline moiety and provide plausible mechanisms for the two-way ring opening of oxazolidine leading to these by-products. In addition, we present evidence that Asi formation appears to be catalyzed by the presence of the pseudoproline moiety. These observed side reactions are attributed to elevated temperature and pressure; therefore, caution is advised when using ΨPro derivatives under such harsh conditions. In addition, we propose a solution whereby thermodynamically controlled Asi formation can be kinetically prevented.

## 1. Introduction

Pseudoproline derivatives such as Ser(Ψ^Me,Me^pro), Thr(Ψ^Me,Me^pro), and Cys(Ψ^Dmp,H^pro) (referred to as Ser(ΨPro), Thr(ΨPro), and Cys(ΨPro), respectively) are commonly used in peptide synthesis, especially for primary sequences with increased aggregation propensities [1,2,3,4,5,6,7] (Figure 1). Similar to proline, these somewhat expensive derivatives are valuable masked residues because they can disrupt the self-aggregation of polypeptide backbones. Thus, the use of Xxx(ΨPro) residues, or ΨPro for short, as building blocks seems crucial in light of current efforts to synthesize longer 30–80 residue polypeptides and protein domains [8,9]. In addition, ΨPro derivatives have the advantage of suppressing aspartimide formation during synthesis by forming a tertiary amine in the chain [10].

Asi is formed most rapidly when the (i + 1) residue with respect to amino acid (i), which is Asp (or Asn), is flexible and has no side chain, making Asp-Gly the most susceptible subunit for isomerization [11,12,13,14,15,16]. Moreover, the catalytic effect of the side chain of the (i + 1) residue has also been proposed, suggesting that Ser and Thr, both residues with short side chains, -Asp-Ser- and -Asp-Thr-, are ready for isomerization within days under synthetic conditions [17]. Therefore, Asi formation and isomerization could be prevented by masking the latter two residues as Ser(ΨPro) or Thr(ΨPro). Due to difficulties in acylating the sterically hindered secondary amine of Xxx(ΨPro) on the resin, ΨPro derivatives are often used in the form of dipeptides Yyy-Xxx(ΨPro) [18]. The use of Xxx(ΨPro) as monomeric building blocks has been the subject of only a few attempts [19,20,21,22]. In light of these considerations, the choice of Xxx(ΨPro) over the expensive Yyy-Xxx(ΨPro) dipeptides would be a desirable approach, using our in-house-developed smart peptide chemistry in flow method or SPF for short [22,23,24] (Figure 2). SPF is not just a device within HPLC modules or just a synthetic protocol; it represents a complex, optimized approach that takes into account synthetic time, reagent quantities, and environmental impact. Therefore, our primary goals are to prioritize reaction efficiency, minimize costs, and promote environmental friendliness. Throughout our work, all syntheses were performed on our continuous flow peptide synthesizer, using previously developed coupling protocols.

In our previous work [22], we screened the coupling efficacy of the Thr(ΨPro) ring using the decapeptide Chignolin005:Y2F (CLN), H-IFDPETGTWI-NH_2_ derived from Honda et al. [25,26]. This relatively short and easily produced oligopeptide was chosen as a model system for protocol optimization, where we used Thr(ΨPro) at the sixth position, and screened all 20 amino acids at the fifth position. Therefore, the 5mer (H-TGTWI-NH_2_) may be a by-product of our synthesis. First, there was the repeated observation of a by-product called 5mer*. Based on mass spectrometry (MS) analysis, we obtained [M+1H]1+ = 616.3 g/mol. This molecular mass is surprisingly ~40 g/mol heavier than the expected molecular weight (MW) (576.30 g/mol) of the truncated sequence H-TGTWI-NH_2_; therefore, it is called H-*TGTWI-NH_2_ or 5mer*. We found that the intact pseudoproline–oxazolidine ring is responsible for the increased MW of this pentapeptide. The possible molecular structure of the 5mer* is therefore H-T(ΨPro)GTWI-NH_2_ (Table 1, Figure 2). From a chemical point of view, the presence of the oxazolidine ring is unexpected, since it is expected to be an acid labile, but it remained intact even after the use of 95% trifluoroacetic acid (TFA) to cleave the polypeptides from the resin [3]. Although the amount of the 5mer* varies from batch to batch, it reaches 2–7% of the crude peptide (Figure 2). This was rather unexpected as no evidence of an acid stable oxazolidine ring contacting peptides has been reported [19,20]. The second unexpected observation was the high amount of Asi formed. During the SPF synthesis of H-IFDPD^5^T^6^GTWI-NH_2_, where the sixth residue was the Thr-masking pseudoproline derivative Thr(ΨPro), we expected that Asi formation would be hindered. However, our observation was the opposite, namely, that the presence of the pseudoproline moiety is a catalyst for Asi formation (Table 1 and Figure 2). In summary, the expected decapeptide CLN is only obtained in 8%, and the unexpected 5mer* is heavier at 40 g/mol [22], indicating that the science behind making this “simple” polypeptide is somewhat different and perhaps more difficult than expected.

**Aim:** We sought to explain how an acid-labile oxazolidine ring can nevertheless withstand treatment with 95% trifluoroacetic acid (TFA) and to understand why Asi formation is catalyzed rather than hindered by pseudoproline amino acid residues.

## 2. Results

### 2.1. Working Hypothesis

To address the challenges outlined above, we designed a series of experiments in which we systematically varied the key parameters of flow peptide synthesis to decipher the stability of the oxazolidine ring and the reason for Asi formation. In addition, we had to rule out any other possible hidden factors that might influence these “side reactions”. First, we decoupled the two problems and focused on the unexpected stability of the 5mer* and then returned to the question of high Asi formation.

### 2.2. The 5mer* Problem

To address the concerns regarding the presence of 5mer*, we conducted a series of six experiments aimed at elucidating the molecular mechanism of the convincing presence of 5mer*. During synthesis #1, we investigated the effect of the Fmoc cleavage from Thr(ΨPro) using our SPF system. Synthesis #2 served as a control for synthesis #1, focusing on the Fmoc protecting group left intact on the Thr(ΨPro) amino acid. In synthesis #3, we aimed to evaluate the effect of SPF and the coupling of the Fmoc-Thr(ΨPro)-OH. Here, we synthesized the 4mer (H-GTWI-NH-resin) using SPF but manually coupled the Fmoc-Thr(ΨPro)-OH residue, followed by the manual cleavage of the Fmoc protecting group. Synthesis #4 replicated the procedures of synthesis #3, except we intended to leave the Fmoc group on the peptide during cleavage. Synthesis #5 was designed to investigate the effect of temperature on the cleavage of the Fmoc group from Thr(ΨPro). Finally, synthesis #6 was designed to evaluate the effect of temperature on the stability of the 5mer in the absence of the pseudoproline derivative, using the simple Fmoc-Thr(*^t^*Bu)-OH amino acid. The results of these experiments were collected (Table 2) and analyzed below. For the corresponding chromatograms, please refer to Appendix A.

### 2.3. The Formation of Asi

One of the most studied side reactions during SPPS is the formation of Asi. Bodanszky et al. [17] presented two pathways to explain this cyclization reaction. Figure 4. The acidic pathway produces a better electrophile from the carboxyl functional group. The base catalyzed pathway produces a better nucleophile from the backbone amide nitrogen. In Fmoc-SPPS, the base-catalyzed model is responsible for most of the Asi formation because piperidine is commonly used as the Fmoc cleavage group. 

We have previously shown that there is an unexpectedly large amount of Asi that is formed during the synthesis of SPF [22].

### 2.4. NMR Study of Asi Form of CLN

First, we wanted to show conclusively that the controversial formation of the Asi derivative was indeed the case when the Asp(tBu)-Thr(ΨPro) was used in the synthesis of CLN. In addition to the MS data, 2D-NMR spectroscopy was used for validation. The peptide and the potential aspartimide derivative were isolated and purified by HPLC prior to NMR analysis. The assignment of both derivatives was completed using the recorded 1H-1H homonuclear correlated spectroscopy (COSY), total correlated spectroscopy (TOCSY) and nuclear Overhauser effect spectroscopy (NOESY) spectra. The absence of the amide proton signal, together with the presence of the remaining observable side-chain protons of Thr at position 6, suggests the formation of the succinimide ring from aspartic acid. (See Section 4 for more information).

### 2.5. Set of Experiments on CLN-Asi Formation 

We used the same model peptide CLN as previously was carried out [22,23] to decipher Asi formation as the challenge was first seen when making it with SPF. Note that this sequence could be a relevant and generalized model, without any special feature. The first synthesis #7 of the CLN served as a “baseline”, focusing on the coupling of Asp to a monomeric Thr(ΨPro) moiety, with the aim of determining the percentage of Asi formation under typical SPF conditions. In synthesis #8, we changed the key residue Thr(ΨPro) to the common Thr(*^t^*Bu) amino acid residue. This synthesis allowed us to observe the effect of the ΨPro ring on this coupling reaction. The synthesis of #9 included the incorporation of Ser(ΨPro) to study the difference between the Ser and Thr. The synthesis of #10 aimed to test the effect of Asp side chain chirality and thus the steric position by substituting L-Asp(*^t^*Bu) with D-Asp(*^t^*Bu). The synthesis of #11 investigated the effect of changing Asp to Asn. Furthermore, the synthesis of #12 focused on the size of the Asp side chain protecting group (*^t^*Bu) when it was replaced with a sterically larger one, namely, that of 5-butyl-nonan (Bno), to see the effect of the bulkiness of the protecting group. The synthesis of #13 was aimed at testing the effect of coupling reagents by switching from acidic ethyl 2-cyano-2-(hydroxyimino)acetate (OxymaPure)/*N,N’*-diisopropylcarbodiimide (DIC) to hexafluorophosphate azabenzotriazole tetramethyl uronium (HATU)/diisopropylethylamine (DIPEA). 

Syntheses #14 and #15 examined the effect of both the coupling time and reagent concentration. These adjustments not only increased the concentration of the coupling reagents but also increased the coupling cycle time. Syntheses of #17 and #18 kept the same parameters used to prepare #7, but the cleavage conditions were modified to see the effects of the different scavengers. In addition, during the synthesis of #18, the role of water molecules as scavengers was tested to detect the products of Asi hydrolysis by using acidic conditions. Similarly, during the synthesis of #19, we hydrolyzed the crude product obtained during the synthesis of #18, but using basic conditions. A comprehensive comparison of the reaction results is summarized in Table 3, where the results clearly reflect the systematic modification of the reaction conditions. For the corresponding chromatograms, please refer to Appendix A.

## 3. Discussion 

### 3.1. Explaining the Appearance of the 5mer* 

We have previously explained that the presence of the 5mer* (MW = 616.3 g/mol) implies the presence of the intact oxazolidine ring, which is somewhat contradictory after the treatment of the crude peptide for 3.5 h with 95% TFA. The results of the 5mer syntheses (#1–#6) lead to the following conclusions: -Synthesis #1 suggests that an elevated temperature (80 °C) favors the formation of 5mer*; this formation can be slowed down (30% to 17%) by leaving the Fmoc group on the 5mer (#2).-The coupling and manual cleavage of Thr(ΨPro) reduces the formation of the 5mer* (30% to 2%) (respectively, #1–#3).-In synthesis #5, it is clearly seen that the Fmoc deprotection is somewhat promoted in the case of the Thr(ΨPro) moiety.-As it was shown in synthesis #6, thermal Fmoc deprotection can also occur in the case of common amino acids.

The only possible explanation for the detectable presence of the oxazolidine moiety came from the synthesis of #1 and #3. The difference between the two syntheses was the coupling and deprotection conditions of the Thr(ΨPro). We found that during the synthesis of #1, the 5mer* (MW= 616.33 g/mol) appeared (Table 2 and Figure 5) by using the SPF method. However, when Thr(ΨPro) (the last amino acid) was manually coupled and the Fmoc group was manually deprotected (#3 at room temperature), we found mostly the 5mer (75%, MW = 576.30 g/mol) and a very small amount of 5mer* (2%, MW = 576.30 + 40 g/mol in Figure 5B). Note that for each crude peptide, the same final cleavage protocol from the resin (95% TFA) was applied. 

Focusing on the major difference, temperature, ~20 °C versus 80 °C, our results suggest that during the Fmoc cleavage, the oxazolidine ring becomes thermally unstable under SPF conditions (80 °C, 6–7 MPa) and forms a stable imine that can withstand the harsh conditions of the acidic (95% TFA) resin cleavage. In addition, the protonated imine form appears to be more stable than the oxazolidine ring [27,28,29,30,31]. 

This mechanism combines two simultaneous reactions (Figure 6). Pathway A is a reaction typical of room temperature, while pathway B illustrates the rearrangement of the oxazolidine ring to its open imine form, which occurs at a high temperature (and elevated pressure). The equilibrium between the open imine form and the oxazolidine form exists, but during the final cleavage from the resin, this equilibrium is frozen as the oxazolidine form is hydrolyzed, but the imine form is protonated, which remains stable at any pH [27]. Interestingly, we calculated the average percentage of 5-mers for all 20 residues as examples to be 4.1% [22]. This means that during the synthesis in our case, the equilibrium constant (Kc) between the open imine and the ring-closed oxazolidine form is Kc = (4.1%)/(95.9%) = 0.0428 [27].

### 3.2. Explaining the Appearance of the CLN-Asi 

From experiments #7–17 described above, the following conclusions were drawn regarding the formation of Asi:-Comparing the Asi by-product ratio of synthesis #7 with that of synthesis #8, we found that the oxazolidine ring of Thr(ΨPro) enhances Asi formation as 30% CLN-Asi is formed compared to the less than 1% observed when Thr(*^t^*Bu) was used.-Synthesis #9 shows that a relatively high Asi formation requires the presence of the oxazolidine ring, Ser(ΨPro) or Thr(ΨPro), but is less sensitive to the -CH_2_-OH or CH(CH_3_)-OH side chain composition of the parent Ser or Thr residue. The observed Asi ratios are 30% and 74%, respectively. This significant difference (30% vs. 74%) can be explained by the more efficient coupling of the Ser(ΨPro) moiety [20,23], resulting in the elimination of the 9-mer in this case. This indicates that although almost all couplings are successful, the resulting peptide is entirely an Asi derivative.-Synthesis #10, with respect to #7, shows that the chirality of the side chain is unimportant since 30% Asi is obtained for the L stereoisomer and 29% for the D stereoisomer.-Synthesis #11 showed that Thr(ΨPro) catalyzes the formation of Asi only for the Asp derivative but not for Asn. The latter gave less than 1% Asi under identical reaction conditions.-Synthesis #12 shows that the steric hindrance of the Asp side chain protecting group is important as Asi formation is greatly reduced when Asp(Bno) is used in addition to Thr(ΨPro).-Syntheses #13–15 show that Asi formation is independent of the coupling reagents, although the length of the coupling cycle and the increased concentration of DIC/OxymaPure somewhat promote extensive Asi formation.-Using the Fmoc-L-Asp(*^t^*Bu)-Thr(ΨPro)-OH dipeptide throughout our synthesis (#16) seems to suppress Asi formation, but we can identify four different products with the same mass.-Varying the final cleavage conditions of the crude polypeptide from resin does not affect Asi formation: it is independent of these conditions (#17 and #18).-However, adding more water to cleavage solution #18 or the basic hydrolysis of crude peptide #19 results in a mixture of peptides with the same mass but different retention times. According to the literature, this suggests that Asi can be hydrolyzed into four different products as D/L-α,β (iso)peptides. However, in our case, we observed at least five different products of the same MW, as shown in Figure 7.

According to the principle of parsimony, the existence of the 5mer* and the above conclusions can also be explained by the instability of the pseudoproline ring. 

However, this mechanism, shown in Figure 8, does not facilitate the formation of the imine moiety. Instead, acetonide deprotection of oxazolidine **A** occurs leaving behind zwitterion **B** (Figure 8). Besides equilibrium racemistaion to **B1**, **B** undergoes sequential amide *trans*→*cis* isomerisation and cyclisation leading to eight-membered cationic activated lactone **C** carrying an amidate moiety. This intermediate is preformed for intramolecular ring contraction by the nucleophilic attack of the amidate nitrogen on the *tert*-butylated lactone carbonyl. The resulting pyrrolidone **D** can undergo cationic tert-butyl migration to give succinimide-centered product **E** (Figure 8). This mechanism also explains the formation of different products with the same molecular weight, as observed for structure B-B1. Cα-H undergoes racemization, resulting in diastereomeric products after hydrolysis, such as Thr(D/L)-Asp(D/L-α,β) peptides. This thermodynamically stable product can be inhibited by kinetic inhibition, as shown in reaction #12, where the bulky Bno protecting group prevents Asi formation even in the presence of the ΨPro ring. The proposed reaction mechanism scenario adequately explains the observed results of the reactions, especially with respect to the detected product composition.

## 4. Materials and Methods

The flow peptide apparatus (HPPS-4000, METALON Ltd., Budapest, Hungary) used is a modified version of a standard Jasco LC-4000 series high-performance liquid chromatography (HPLC) system. The modification involves the addition of an extra valve to the PU-4180 HPLC pump and the control of solvent flow, such as the cleavage mixture used in peptide synthesis. ChromNAV2, software version 2.04.06 was used to fully automate the process. Reagent injection was performed by an autosampler from 2 mL sample vials, placed in the sample rack. The column made of polyetheretherketone (PEEK) column material was used as a fixed-bed reactor tube for the resin. Dimethylformamide (DMF) was used as the solvent and eluent. For all syntheses, ~150 mg of Fmoc-Rink amide TentaGel resin (0.24 mmol/g) was loaded into the PEEK column. All peptides were synthesized using the Fmoc/*^t^*Bu strategy. Tentagel S resin was purchased from Rapp Polymere GmbH (Tübingen, Germany), and the Fmoc-Ser(Ψ^Me,Me^Pro)-OH, Fmoc-Thr(Ψ^Me,Me^Pro)-OH, and Fmoc-Asp(O*^t^*Bu)-Thr(Ψ^Me,Me^Pro)-OH were purchased from Iris Biotech GmbH (Marktredwitz, Germany). 

The reagent solutions were injected onto the resin-filled column. For Fmoc deprotection, the deprotection solution consisted of 30 *v*/*v*% piperidine in DMF. DMF was used for the washing steps during the synthesis. In the vials, the protected amino acids were dissolved with coupling reagents *N*-methyl-2-pyrrolidone (NMP). The activating agent (DIC) was added just before the coupling and was injected with the autosampler. During the synthesis, the pressure varied between 5 and 6 MPa, using a back-pressure regulator. 

After peptide synthesis, the resin was washed with dichloromethane (DCM) and then dried under vacuum. Cleavage from the resin was performed with (A) TFA: H_2_O:TIS/95:2.5:2.5 (*v*/*v* %) or (B) TFA:H_2_O:thioanisole:EDT:TIS:phenol/5 mL:250 μL:250 μL:125 μL:60 μL:250 mg) with stirring for ~3.5 h at room temperature. The solution was then filtered, and TFA was removed from the solution using a rotary vacuum evaporator. The peptide was washed with diethyl ether and dried under vacuum, solubilized in water or water/acetonitrile solution, and lyophilized.

### 4.1. UPLC-MS Conditions 

The cleaved products of syntheses #7–#19 were analyzed by ultra-performance liquid chromatography–mass spectrometry (UPLC-MS) on an ACQUITY UPLC^®^ HSS T3 1.8µm column (Waters, 100 mm × 2.1 mm, 100 Å) using a gradient elution consisting of 0.1% TFA in water (eluent A) and 0.1% TFA in acetonitrile (eluent B) with a gradient from 25% to 65% B over 8 min and the column temperature set at 45 °C. The flow rate was set to 0.18 mL/min, and the absorbance was detected at λ = 220 nm. LC-MS analysis of the compounds was performed on Waters Select Series Cyclic IMS MS connected directly to Waters Acquity I-Class Plus UPLC. The data were analyzed using Waters MassLynx V4.2 (Waters Corporation, Milford, UK). The crude products of syntheses #1–#6 were analyzed by MS: Waters ACQUITY RDa Detector and on UPLC: Waters ACQUITY UPLC H-Class PLUS System using UNIFI 1.9.13.9 (Waters Corporation, Milford, UK).

### 4.2. Basic Hydrolysis of CLN

Moreover, 10 mg of CLN peptide was dissolved in 2 mL of NaOH solution (0.01 M) for 24 h. After lyophilization, the crude peptide weight was 6.9 mg.

### 4.3. Purification of Asi and CLN Peptide 

The crude peptide resulting from synthesis #7 (0.42 mg) was dissolved in 18 mL of water. The purification of the peptide resulting from synthesis #7 was performed on an HPLC (Jasco Corporation, Tokyo, Japan) using a Kinetex 5μm C18 100 Å 250 mm × 10 mm column. We used a gradient elution consisting of 0.1% TFA in water (A) and 0.08% TFA in acetonitrile (B) with a gradient from 10% B to 60% B over 120 min. The absorbance was detected at 220 nm. The flowrate was 4.5 mL/min. The same retention time fractions were collected and, after lyophilization, yielded 1.073 mg of CLN-Asi peptide and 0.47 mg of pure CLN. The NMR measurements were performed on these purified peptides.

### 4.4. NMR Analysis

Aqueous solutions of lyophilized CLN-Asi containing and pure CLN-s were prepared at a concentration of 0.8 mM, with a total volume ranging from 500 to 700 μL. These solutions contained 10% D_2_O and 1% NaN_3_. The pH of the samples was adjusted to 4.3 using 0.1 M NaOH and 0.1 M HCl. NMR measurements were performed on a 16.4T (700 MHz) Bruker Avance III spectrometer, equipped with a Prodigy TCI H&F-C/N-D probe at 288K. Full proton resonance assignments were carried out utilizing water-suppressed 2D ^1^H-^1^H homonuclear DQF-COSY (cosygpprqf), TOCSY (mlevgpph19), and NOESY (noesygpph19) standard Bruker experiments. The resolutions were set to 2048 × 1024, with 124 scans. For TOCSY measurements, a spinlock of d9 = 80 ms was applied, and for NOESY measurements, a mixing time of d8 = 150 ms was used. Spectrum processing was performed using the Topspin software, version 4.0.7. Spectra assignment was carried out using the CCPNMR software, version 2.4.1.51. For the designated region under discussion and the corresponding NMR shift list, please refer to Appendix A.

## 5. Conclusions

In conclusion, we have identified the Achilles’ heel of the ΨPro ring under SPF conditions. The appearance of two unexpected by-products, namely, the imine form of 5mer* and the increased formation of CLN-Asi, can be attributed to the oxazolidine character of the pseudoproline ring. The ring opening of ΨPro is favored at high temperatures and elevated pressures (80 °C, 5–6 MPa). However, Asi formation is thermodynamically favored at high temperatures. It has been shown that this reaction can be kinetically hindered by using bulkier protecting groups such as Bno on the Asp side chain. Given the importance of elevated temperature [32,33] and the utility of ΨPro in polypeptide synthesis, caution is required in its use. This study not only sheds light on peptide chemistry but also contributes to the literature on oxazolidine rings.

## Figures and Tables

**Figure 1 ijms-25-04150-f001:**
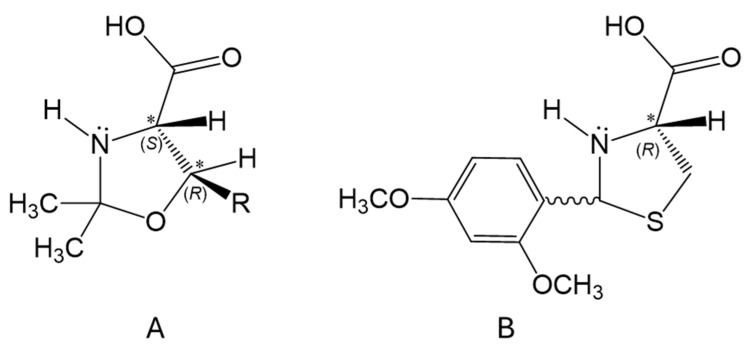
The chemical structure of (**A**) H-l-Ser(Ψ^Me,Me^pro)-OH (R = H), H-l-Thr(Ψ^Me,Me^pro)-OH (R = CH_3_), and (**B**) H-l-Cys(Ψ^Dmp,H^pro)-OH (Dmp: 2,4-dimethoxphenyl).

**Figure 2 ijms-25-04150-f002:**
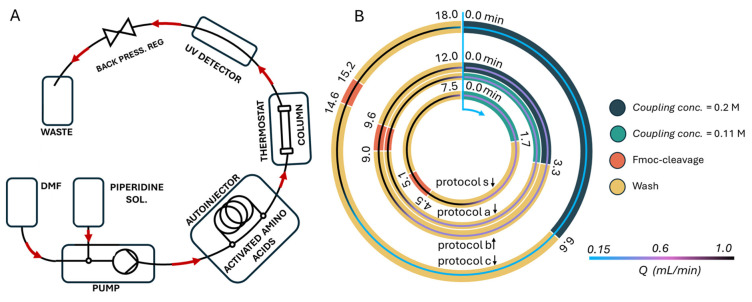
(**A**) Schematic diagram of the smart peptide chemistry in flow apparatus (HPPS-4000, METALON Ltd., Budapest, Hungary). (**B**) Flow diagrams for coupling protocols s, a, b, and c are provided, with arrows indicating the corresponding circle, respectively. The fastest is protocol s, which is used to couple all amino acid derivatives except ΨPro. The inner thin light blue, purple, and black circles represent the flow rate Q in (mL/min). A more detailed tabular version of the synthetic protocols (**B**) is included in Appendix A.

**Figure 3 ijms-25-04150-f003:**
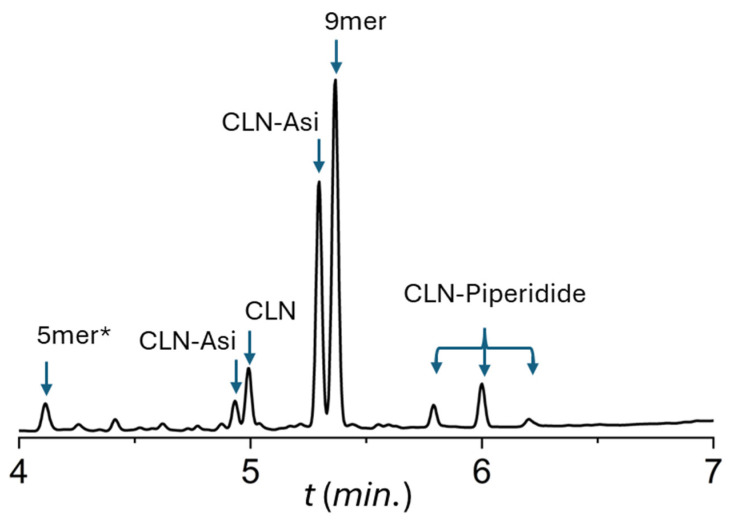
Ultra-performance liquid chromatography (UPLC) chromatogram of the crude CLN decapeptide in which the peaks have been identified via mass spectrometry [22]. During the synthesis of CLN, both truncated sequences (e.g., 9mer) and chemically modified by-products (e.g., 5mer*, CLN-Asi, CLN-Piperidide) are formed. As shown, two major CLN-Asi and three major CLN-piperidides are formed (e.g., α/β-D/L CLN-Asi and CLN-piperidide).

**Figure 4 ijms-25-04150-f004:**
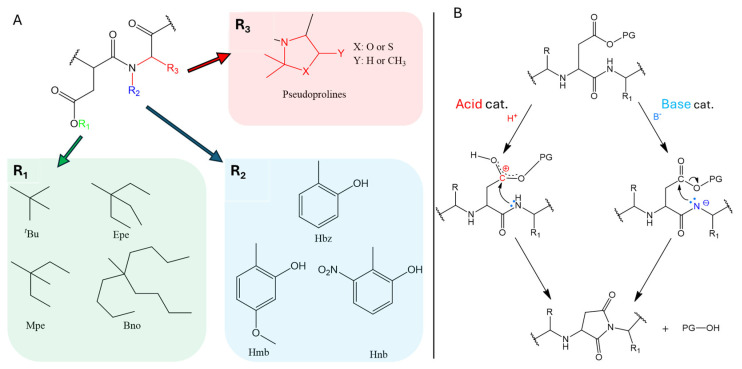
(**A**) Different strategies to prevent Asi formation by using different protecting groups. R1: bulky ester protecting the side chain of Asp (e.g: 3-ethyl-3-pentyl (Epe), 3-methyl-3-pentyl (Mpe), 5-butyl-5-nonyl (Bno). R2: an aromatic group protects the NαGly (e.g., 2-hydroxy-benzole (Hbz), 2-hydroxy-3-methoxy-benzole (Hmb), 2-hydroxy-5-nitro-benzole (Hnb)). R2 and R3: pseudoproline derivatives are used when the amino acid Thr, Ser or Cys is present instead of Gly within the sequence at the (i + 1) position “following” Asp [6]. (**B**) The two ways of aspartimide formation during peptide synthesis reported by Bodansky et al. [17].

**Figure 5 ijms-25-04150-f005:**
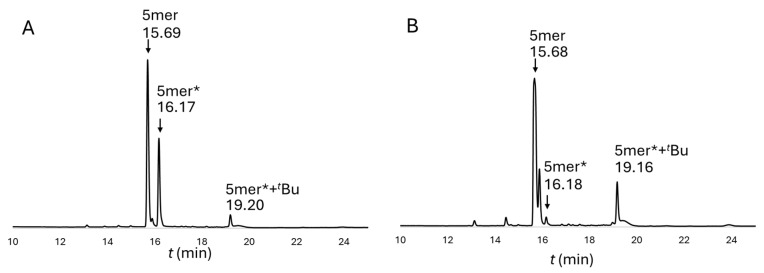
HPLC chromatograms of the crude peptides during (**A**) synthesis condition #1 and (**B**) synthetic condition #3. Note that synthetic condition #3 includes the manual and room temperature coupling of the critical Thr(ΨPro) residue. The chromatograms of syntheses #2, #4, #5, and #6 are shown in Appendix A, respectively.

**Figure 6 ijms-25-04150-f006:**
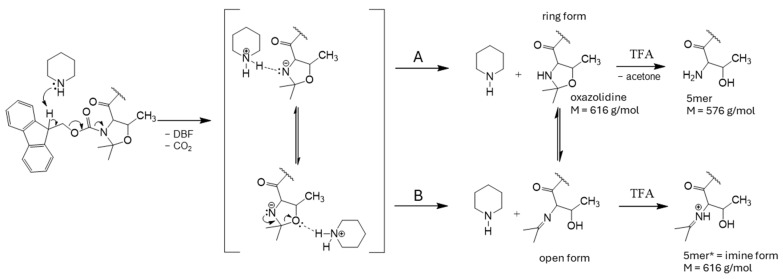
The detailed molecular mechanism of the formation of the by-product 5mer through path A and 5mer*, along which the open imine is formed from the oxazolidine ring via reaction pathway B.

**Figure 7 ijms-25-04150-f007:**
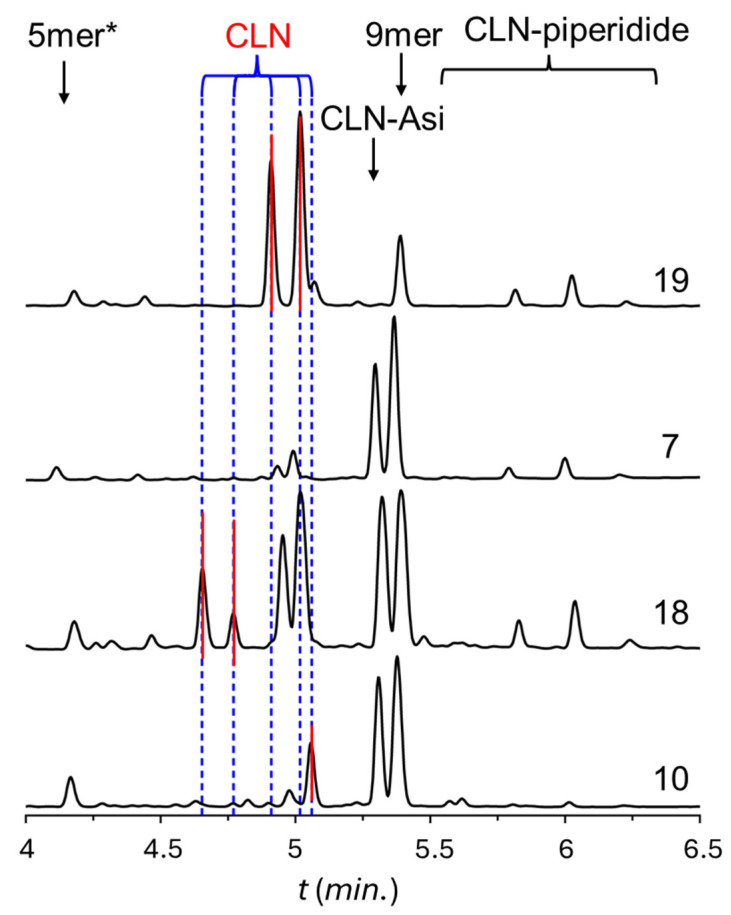
The UPLC chromatograms of the raw product are obtained from four different reactions (19, 7, 18, and 10). We assigned five different oligopeptides (peaks, marked with red lines) of identical *m*/*z* = 1163.5: products explained by the formation of the D/L- and α, β-isomers.

**Figure 8 ijms-25-04150-f008:**
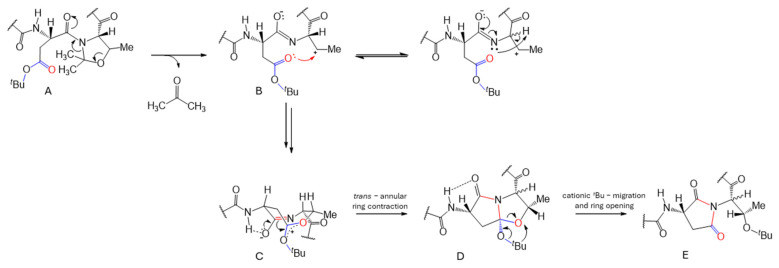
Possible mechanism of Asi formation during the thermal instability of the ΨPro ring.

**Table 1 ijms-25-04150-t001:** Molecular masses and retention times of the CLN decapeptide and its by-products during SPF synthesis (see Figure 3 for the UPLC chromatogram).

Name	Sequence	Retention Time (*min*)	Mass [M+H]^+^ (*m*/*z*)
CLN	H-IFDPDT GTWI-NH_2_	4.99	1163.56
9mer	H-IFDPT GTWI-NH_2_	5.37	1048.54
5mer*	H-*TGTWI-NH_2_	4.11	616.34
CLN-Asi	H-IFDPD^Asi^T GTWI-NH_2_	4.93 + 5.30	1145.55
CLN-Piperidide	H-IFDPD^Pip^T GTWI-NH_2_	5.79 + 6.00 + 6.20	1230.64

**Table 2 ijms-25-04150-t002:** Formation of both the 5mer (H-TGTWI-NH_2_) and 5mer* (H-*TGTWI-NH_2_) during the different synthetic conditions from N#1 to N#6.

N#	5mer %	5mer* %	5mer-Fmoc %	Synthetic Cond. ^a^	Reaction Cond. ^b^
1	62	30	-	SPF	80 °C/2 h
2	60	17	2/2	SPF	80 °C/2 h
3	75	2	-	Manual	r.t./2 h
4	-	-	>99	Manual	r.t./2 h
5	88	<1	<1	SPF	r.t./2 h
6	>95	-	<1	SPF	80 °C/2 h

^a^ All amino acids were coupled using our in-house-developed SPF method and apparatus. However, the last *N*-terminal amino acids were varied [22,23,24]. ^b^ Conditions used after coupling the Fmoc-Thr(ΨPro)-OH (#1–#5) or Fmoc-Thr(*^t^*Bu)-OH (#6) residue to see the effect of temperature after coupling.

**Table 3 ijms-25-04150-t003:** Selected data to decipher the origin of the 2 significant by-products, Asi and the 9mer, during the synthesis of H-IFDPD*T(ΨPro)*GTWI-NH_2_ using the different synthetic conditions from N#7 to N#19.

N#	Variations in Sequence	CLN. %	Asi %	9mer %	5mer* %	Synthetic Protocols ^b^	Cleavage Cond. ^c^
7	H-IFDPD***T(ΨPro) ^a^*** GTWI-NH_2_	8	30	45	4	a	A
8	H-IFDPD***T(**^t^***Bu*)***GTWI-NH_2_	86	<1	<1	-	a	A
9	H-IFDPD***S(ΨPro)***GTWI-NH_2_	-	74 ^d^	-	4	a	A
10	H-IFDP***d**T(ΨPro)*GTWI-NH_2_	14	29	38	7	a	A
11	H-IFDP***N**T(ΨPro)*GTWI-NH_2_	78	<1	~1	3	a	A
12	H-IFDP***D(Bno)T(ΨPro)***GTWI-NH_2_	73	<1	16	4	a	A
13	H-IFDPD*T(ΨPro)*GTWI-NH_2_	10	19	28	5	**a (HATU/DIPEA)**	A
14	H-IFDPD*T(ΨPro)*GTWI-NH_2_	8	63 ^d^	<1	3	**b**	A
15	H-IFDPD*T(ΨPro)*GTWI-NH_2_	8	65 ^d^	<1	6	**c**	A
16	H-IFDP**D*T(ΨPro)***GTWI-NH_2_	78 ^d^	-	-	-	a	A
17	H-IFDPD*T(ΨPro)*GTWI-NH_2_	11	21	40	7	a	**B**
18	H-IFDPD*T(ΨPro)*GTWI-NH_2_	31 ^d^	50 ^d^	-	3	a	**C**
19	H-IFDPD*T(ΨPro)*GTWI-NH_2_	69 ^d^	-	13	3	a	A ^e^

^***a***^ Variable amino acid parameters in bold; ^b^ for the synthetic protocols, see Figure 2; ^c^ cleavage protocols are as follows: A) TFA: H_2_O: Triisopropylsilane (TIS)/95:2.5:2.5 (*v*/*v* %); B) TFA: H_2_O: thioanisole: 1,2-ethanedithiol (EDT): TIS: phenol/5 mL: 250 μL: 250 μL: 125 μL: 60 μL: 250 mg; C) TFA: H_2_O: TIS/77:18:5 (*v*/*v* %); ^d^ sum of several peptides with the same mass but different retention time (for more details, see the Appendix A). ^e^ The crude peptide was treated with a solution of 0.01 M NaOH.

## Data Availability

Data is contained within the article and Appendix A.

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
