# Peer review of "Unveiling the Oxazolidine Character of Pseudoproline Derivatives by Automated Flow Peptide Chemistry"

_ijms, 2024, doi:10.3390/ijms25084150_

Round 1
Reviewer 1 Report
Comments and Suggestions for Authors
This study focused on the investigation of byproducts formed during the synthesis of decapeptide using Thr(ΨPro) and suggested a mechanism for the two-way opening of the oxazolidine ring and Asi formation.
Overall, this manuscript does not read well due to a lack of details and clarifications that need to be addressed, properly. The details are in the comments in the attached text file with some suggestions for improvement:
- Since the study is about peptide bond formation, the term “acylation” is better replaced with “coupling” or other terms specific to peptide synthesis, as “acylation” refers to a broad class of chemical reactions.
- Be mindful of the unit “Da”, which is typically used for proteins with high molecular weight.
- The reviewer is concerned about the high-temperature running of SPF. In general, peptide syntheses are performed at lower temperatures than 50 degrees for multiple reasons. Unless synthesis is performed at high temperatures, the problems pointed out in this study are unlikely to occur. Therefore, the authors should further state the take-home message.
- It seems necessary to confirm the racemized byproducts, Thr(D/L)-Asp(D/L-α,β) peptides, by synthesizing authentic peptides for comparison.
- Ensure consistent spacing between a number and a unit, except for percentage.
- The reviewer suggests adding error (ppm) to mass values in the supplementary file.
Additionally, the reviewer recommends having the manuscript proofread by an English expert to improve readability. Once these issues are addressed, your manuscript may be suitable for publication in this journal.

Additionally, the reviewer recommends having the manuscript proofread by an English expert to improve readability.
Reviewer 2 Report
Comments and Suggestions for Authors
Szaniszló and co-workers reported a study on the oxazolidine character of pseudoproline derivatives by automated flow peptide chemistry. The study involved the design of a series of controlled experiments and some mechanistic sequences proposed in order to explain the formation of the expected products as well as several by-products detected and characterized by analytical/spectroscopic techniques. Practical solutions were also supplied for some synthetic drawbacks in this process.
Some minor revisions are recommended as follow:
- Several style/grammatical corrections must be undertaken into the document. Please see the attached file.
- Figure 3 and some inconsistences observed in the mechanistic proposals drawn in Figures 5 and 7 should be revised and corrected as suggested in the attached file.
- The references style must be revised, unified and corrected. They are too mixed.

Round 2
Reviewer 1 Report
Comments and Suggestions for Authors
The manuscript has been improved after revisions to the issues raised by the reviewer, and the reviewer consider it acceptable for the publication.
Comments on the Quality of English LanguageSince there are still some parts doesn't read well, the reviewer suggest the authors having proof-reading from an English expert.
Author Response
Dear Reviewer,
Thank you for your valuable feedback. We appreciate your thorough review of our manuscript.
Best regards,
Szebasztián Szaniszló